# Coating Doyle Nasal Silicone Splints with a Sustained Release Varnish Containing Antibiotics Provides Long-Term Protection from *Staphylococcus aureus*: An In Vitro Study

**DOI:** 10.3390/ph18111746

**Published:** 2025-11-17

**Authors:** Ahmad Siag, Ronit Vogt Sionov, Irith Gati, Michael Friedman, Doron Steinberg, Menachem Gross

**Affiliations:** 1Institute of Biomedical and Oral Research (IBOR), Faculty of Dental Medicine, The Hebrew University of Jerusalem, Ein Kerem Campus, Jerusalem 9112102, Israel; ahmadsi@hadassah.org.il (A.S.); ronit.sionov@mail.huji.ac.il (R.V.S.); dorons@ekmd.huji.ac.il (D.S.); 2Department of Otolaryngology-Head and Neck Surgery, Hadassah Medical Center, Jerusalem 9112102, Israel; 3The Institute of Drug Research, School of Pharmacy, The Hebrew University of Jerusalem, Jerusalem 9112102, Israel; irith.gati@mail.huji.ac.il (I.G.); michaelf@ekmd.huji.ac.il (M.F.)

**Keywords:** Doyle nasal silicone splints, sustained-release varnish (SRV), augmentin, ciprofloxacin, chloramphenicol, *Staphylococcus aureus*

## Abstract

**Background/Objectives**: Doyle nasal silicone splints are commonly used in nasal surgeries to maintain the shape of the nasal passage and prevent scar tissue formation. However, these implants are prone to bacterial colonization, particularly by *Staphylococcus aureus*, which is associated with severely recurrent and recalcitrant cases of infected sinonasal cavities. The aim of this study was to develop a sustained-release varnish (SRV) with antibacterial properties that can be applied to Doyle splints to provide an antibacterial environment for an extended period. **Methods**: Doyle nasal splints (1 cm × 1 cm segments) were coated with SRV containing one of the three antibiotics: augmentin, ciprofloxacin, or chloramphenicol. A placebo varnish without antibiotics served as a control. The coated splints were exposed daily to a fresh culture of *S. aureus*, and antibacterial activity was assessed by monitoring bacterial growth. Antibiofilm activity was determined using an MTT metabolic assay. Antibacterial activity was further studied by the kinetic disk diffusion assay, where the stents were transferred daily to new, freshly coated *S. aureus* plates. Biofilm formation on the coated splints was visualized by high-resolution scanning electron microscopy (HR-SEM). **Results**: Doyle segments coated with augmentin, ciprofloxacin, or chloramphenicol effectively inhibited *S. aureus* planktonic growth for 9 ± 1, 18 ± 1, and 21 ± 1 days, respectively. Biofilm formation was prevented for 10 ± 1, 18 ± 1, and 21 ± 1 days, and bacterial clearance occurred for 14 ± 1, 52 ± 1, and >65 days, respectively. HR-SEM images showed the prevention of biofilm formation on the coated segments. **Conclusions**: Our findings demonstrate that coating Doyle nasal silicon splints with SRV containing augmentin, ciprofloxacin, or chloramphenicol provides long-term antibacterial and antibiofilm activity, with SRV–chloramphenicol being superior. Further studies are needed to confirm the in vivo efficacy of this approach.

## 1. Introduction

Endoscopic sinus surgery (ESS) is a commonly performed procedure for patients with recurrent or chronic rhinosinusitis who do not respond to conventional medical therapy [1,2]. One of the goals of ESS is to widen the sinus ostia to improve sinus drainage and prevent future sinus infections. However, some patients continue to have persistent and refractory disease despite surgical treatment, with up to 10% requiring revision surgery within 3 years [3], and even higher rates (up to 20%) for patients with nasal polyps [4]. The most common complications of ESS that may necessitate revision surgery are lateralization of the middle turbinate (MT) and synechiae formation [5,6]. Intranasal adhesions (synechiae) develop as a result of improper healing of the nasal mucosa, with an incidence ranging from 6.8% to 36% of rhinosurgical procedures [7]. These adhesions can obstruct the normal mucociliary drainage pathway of the sinuses, leading to blockage of the ostiomeatal complex and failure of the procedure [8,9].

Nasal septoplasty is one of the most commonly performed operations in otorhinolaryngology and is often conducted in conjunction with ESS. Nasal packing is routinely used after septoplasty to control bleeding, prevent septal hematoma, and reduce the risk of nasal synechiae. Nasal packing also plays a crucial role in ensuring mucoperichondrial flap coaptation and cartilage stabilization required for optimal surgical results [10,11]. Intranasal stenting is commonly performed with Doyle silicone splints in septoplasty operations [12,13].

Nasal cavity obstruction due to adhesions and synechiae increases the risk of bacterial growth, even in patients with no clinical evidence of infection of the sinuses. The presence of microbial biofilms in patients with chronic sinusitis explains the lack of efficiency of antibiotic treatment at the concentrations typically used in clinical practice [14]. In the United States, *Staphylococcus aureus*, *Pseudomonas aeruginosa*, and *Haemophilus influenzae* are the most common pathogens that form biofilms in the nasal cavity and sinuses [15]. Biofilms are highly organized microbial communities characterized by extreme resistance to antibiotics, host immune defenses, and various chemical and physical agents [16,17,18]. The extracellular matrix surrounding the bacteria in biofilms protects them from host immune defense mechanisms and hinders the penetration of antimicrobial agents. Furthermore, within a biofilm, bacteria exhibit reduced antibiotic susceptibility due to the adaptation of heritable resistance mechanisms, including adaptive mutations and horizontal gene transfer [17,18]. Consequently, the eradication of biofilms using traditional antibiotics is usually not efficient.

Several therapeutic strategies have been explored to combat biofilm-associated infections. High-dose or combination antibiotic therapy at supratherapeutic concentrations of certain antibiotics (e.g., daptomycin, levofloxacin with rifampin) has been shown to be effective in experimental models. However, these high concentrations are typically not achievable in the human body without causing toxicity. Rifampin, in particular, is frequently used in combination therapy for device-associated biofilm infections, though its effectiveness can vary depending on the bacterial strain and the synergy between antibiotics [19,20,21]. Other potential approaches include matrix-disrupting agents to break down the biofilm’s protective structure [22,23], antimicrobial peptides [24], and physical or adjunctive therapies such as photodynamic therapy (PDT), photothermal therapy, ultrasound, and hyperbaric oxygen, which can enhance the effectiveness of antibiotics or directly target biofilm disruption [20,22,25]. Bacteriophage therapy is another avenue being explored, as bacteriophages can selectively target bacterial strains [26,27]. However, this approach tends to be strain-specific, limiting its general applicability. A multidisciplinary and multi-targeted approach is generally required for effective biofilm eradication, especially in device-associated or chronic infections.

The use of topical antibiotics has long been a research focus because they can be delivered at significantly higher concentrations directly to the sinonasal mucosal surface with minimal systemic absorption [28,29]. A study by Ha et al. [30] in which different *S. aureus* strains were allowed to form biofilms in vitro before exposure to increasing concentrations of topical antimicrobials, found that topical mupirocin could reduce *S. aureus* biofilm mass by more than 90%, while topical ciprofloxacin and vancomycin were largely ineffective within the safe dosage range. Given the challenges of eradicating established biofilms, we propose developing a prophylactic strategy to prevent biofilm formation following endoscopic sinus and septoplasty surgeries.

In this research, we developed a novel platform that uses a sustained release varnish (SRV) incorporating various antibiotics such as augmentin, ciprofloxacin, and chloramphenicol as a prophylactic coating on Doyle nasal silicon splints. This method aims to prevent bacterial infection in the nasal cavity and address early post-operative complications, including inflammation, infection, and biofilm formation. The SRV technology provides a localized, sustained release of antibiotics directly to the nasal cavity and sinuses, thereby addressing primary complications without the need for systemic antibiotic administration. This approach not only reduces the risk of infection but also minimizes potential side effects associated with oral or systemic antibiotics. The choice of these three antibiotics was based on their established efficacy in treating rhinosinusitis and their ability to target common pathogens involved in these infections [31,32,33]. Our study is a preclinical in vitro investigation, and future research is required to support the transition of this platform into clinical use.

SRV technology has previously been investigated in vitro in various applications aimed at preventing microbial infections through the incorporation of different antiseptic and antimicrobial compounds, including chlorhexidine [34,35], triclosan [36], ciprofloxacin [37], and clotrimazole [38]. In vivo studies have further demonstrated the clinical potential of this approach: Coating catheters with SRV-chlorhexidine reduced catheter-associated biofilms in dogs [39], while application of SRV-clotrimazole in human volunteers decreased *Candida albicans* load in the oral cavity [40].

Our platform offers several significant advantages over traditional systemic treatments [41,42]. By delivering antibiotics directly to the site of potential infection, we aim to reduce the incidence of post-operative infections, minimize local inflammation, and prevent bacterial colonization in the sinonasal cavity, which can lead to biofilm formation. This targeted approach enhances patient compliance by eliminating the need for frequent administration of systemic antibiotics, which are associated with increased risks of side effects and antibiotic resistance. Additionally, the sustained local release of antibiotics ensures that effective antimicrobial concentrations are maintained over time, reducing overall exposure to antibiotics. This strategy improves patient compliance by offering a more convenient and efficient treatment option, ultimately improving outcomes and reducing the likelihood of post-operative complications.

## 2. Results

The aim of this study was to develop a varnish that can be used to coat Doyle nasal silicone splints with long-term antibacterial and anti-biofilm properties. We hypothesized that incorporating antibiotics commonly used in the treatment of rhinosinusitis into a sustained-release varnish (SRV) could offer significant clinical benefits by preventing post-operative complications, such as infection and biofilm formation.

### 2.1. Coating Doyle Nasal Silicone Splints with SRV Containing Antibiotics Caused Long-Term Inhibition of S. aureus Planktonic Growth and Biofilm Formation

To study whether coating Doyle splints with SRV containing antibiotics is effective in preventing *S. aureus* bacterial growth and biofilm formation, SRV-coated splints (1 cm × 1 cm) were incubated daily with fresh cultures of *S. aureus* ATCC 25923 for 24 h. The effect on planktonic bacterial growth was monitored daily by measuring the optical density (OD) at 600 nm, and the extent of biofilm formation was assessed using the MTT metabolic assay. Our findings show that Doyle splints coated with SRV–augmentin, SRV- ciprofloxacin, or SRV–chloramphenicol significantly prevented *S. aureus* bacterial growth for 9 ± 1, 18 ± 1, and 21 ± 1 days, and *S. aureus* biofilm formation for 10 ± 1, 18 ± 1, and 21 ± 1 days, respectively (Figure 1 and Figure 2). As expected, SRV–placebo-coated splints had no significant antibacterial or antibiofilm effect. Our assumption was that if the amount of antimicrobial agent released from the coatings was sufficient to inhibit bacterial growth and biofilm formation in the surrounding medium, it would likewise be sufficient to prevent bacterial colonization and biofilm formation on the coated segments themselves, where the local concentration of the compound is even higher. The observed decrease in planktonic growth and biofilm formation by SRV–placebo-coated segments may reflect a physical or surface-related interference effect arising from the presence of the segments themselves, rather than from any direct antimicrobial activity.

### 2.2. Coating Doyle Nasal Silicone Splints with SRV Containing Antibiotics Resulted in Long-Term Inhibition of S. aureus Growth in Disk Diffusion Assay

The antibacterial activity of SRV–antibiotic-coated Doyle splints (1 cm × 1 cm) was further evaluated by placing them daily on TSA plates pre-inoculated with *S. aureus*. The area of the inhibition zone around each sample was measured after overnight incubation at 37 °C. The kinetic disk diffusion assay revealed that, at a size of 1 cm × 1 cm, SRV–chloramphenicol-coated Doyle splints delayed *S. aureus* growth for more than 65 days, SRV–ciprofloxacin for 52 ± 1 days, and SRV–augmentin for 14 ± 1 days (Figure 3). SRV–placebo-coated splints showed no inhibitory effect.

### 2.3. HR-SEM Images Showed Significant Reduction of Biofilm Formation on Doyle Nasal Silicone Splints Coated with SRV–Antibiotics

It was also important to determine whether coating Doyle splints with SRV–antibiotics would prevent biofilm formation on the splints. To this end, the coated splints were exposed ten times to *S. aureus*, and the presence of bacterial biofilm was visualized by HR-SEM. The HR-SEM images show strong inhibition of biofilm formation on splints coated with SRV–augmentin, SRV–chloramphenicol or SRV–ciprofloxacin, which was in sharp contrast to the SRV–placebo-coated splints, which were covered by *S. aureus* biofilms (Figure 4). Altogether, our findings show that coating Doyle splints with an SRV containing augmentin, ciprofloxacin, or chloramphenicol provides long-term protection against *S. aureus* bacterial growth and subsequent biofilm formation. Among the three SRV–antibiotics, SRV–chloramphenicol-coated splits had the longest antibacterial and antibiofilm properties.

## 3. Discussion

Doyle nasal silicone splints are intranasal devices commonly used after septal surgery, such as septoplasty or septorhinoplasty, to stabilize the septum, prevent postoperative septal hematoma, and reduce the risk of mucosal adhesions (synechiae) between the septum and the lateral nasal wall. These splints are typically made from medical-grade silicone, with some designs featuring an integral airway to facilitate postoperative nasal breathing [43,44,45].

Clinical studies support their efficacy in maintaining septal alignment and minimizing synechiae formation, although their use has been linked to increased short-term postoperative discomfort, infection risk, and nasal obstruction compared to no-splint approaches. These adverse effects generally resolve after removal, with long-term mucosal healing outcomes remaining comparable between splinted and non-splinted patients [46,47,48]. Splints are usually placed bilaterally for 2–7 days, depending on patient factors, and modifications such as hemi-split splints may reduce removal-related discomfort [49]. Coating splints with topical antibiotics or ointments has been suggested to further reduce colonization and infection [45,50,51].

Despite their advantages, Doyle splints remain foreign bodies and are therefore highly susceptible to biofilm colonization, which contributes to chronic infection, antibiotic resistance, and device failure [52]. The present study demonstrates that coating Doyle splints with SRV–augmentin, SRV–ciprofloxacin, or SRV–chloramphenicol markedly reduces bacterial proliferation and biofilm formation, with SRV–chloramphenicol showing the strongest and most prolonged effect. This finding is clinically relevant, as it highlights a localized strategy that minimizes the need for systemic antibiotics.

Among the tested agents, chloramphenicol’s superior performance can be attributed to its stability within the varnish and its well-established topical safety. Ciprofloxacin also demonstrated prolonged efficacy but was slightly less durable than chloramphenicol, while augmentin’s shorter action period may be due to its low stability in varnish formulations. These results emphasize that the success of SRV systems depends not only on the intrinsic antibacterial spectrum of the drug, but also on its physicochemical compatibility with the sustained-release matrix.

Importantly, sustained local delivery offers several advantages over current postoperative care. Doyle splints are usually left in place for 3–7 days, yet biofilm formation has been documented as early as 48 h, with universal colonization by 96 h [52]. Although there is no clinical consensus on the ideal removal time [46,53], extended use beyond 7 days is rare due to infection risk. By integrating an antibiotic-eluting varnish, the splints become active devices, providing protection well beyond conventional timelines and reducing reliance on oral antibiotics.

The results of the kinetic disk diffusion assay show that SRV–ciprofloxacin and SRV–chloramphenicol coatings on nasal splints can sustain antibacterial activity for approximately two months, while SRV–augmentin maintained activity for only two weeks. Planktonic growth assays and biofilm metabolic tests demonstrated shorter durations (9–21 days, depending on the varnish) than the disk diffusion assay, possibly due to prolonged immersion in fluid. In this respect, the disk diffusion assay more closely mimics the nasal cavity, which has an air-tissue interface. Notably, HR-SEM images provided direct visualization of reduced biofilm accumulation, with SRV-coated devices showing scattered, morphologically abnormal bacteria compared to dense, structured biofilms on placebo splints. Collectively, these data confirm the coatings’ ability to prevent both planktonic growth and biofilm formation.

While the results of the present investigation are significantly promising, several limitations warrant consideration. First, the study was conducted under in vitro conditions, which cannot fully recapitulate the anatomical and physiological complexity of the human nasal cavity. Therefore, further in vivo studies are essential to confirm the clinical efficacy of SRV–antibiotic-coated splints. Second, experimental design does not accurately reflect the physiological environment of the sinonasal cavity. For example, the fluid volume used in the experiments substantially exceeded that typically present in vivo. Another limitation concerns the microbial conditions: In the nasal cavity, SRV–antibiotic-coated splints are expected to eliminate most bacteria during the early period after insertion, resulting in an almost sterile environment. This contrasts with our model, in which fresh bacterial suspension was replenished daily.

Overall, our results establish SRV-coated Doyle splints as a promising tool for reducing biofilm formation, enhancing infection control, and minimizing systemic antibiotic exposure in otorhinolaryngology.

## 4. Materials and Methods

### 4.1. Preparation of Sustained Release Varnish (SRV)

The SRVs were prepared by modifying the method described by Sionov et al. [38] with specific pharmaceutical adaptations to ensure adhesion to silicone surfaces. Polyethylene glycol 400 (PEG-400, Sigma, St. Louis, MO, USA) was dissolved in ethanol, followed by the addition of the polymeric components ethylcellulose (Sigma, St. Louis, MO, USA) and hydroxypropylcellulose (Sigma, St. Louis, MO, USA), which were allowed to dissolve completely through continuous mixing at 150 rpm for 24 h at 37 °C. After cooling the solution to room temperature, the active agents were incorporated into the formulations (Table 1). The antibacterial drugs used were augmentin (BioAvenir—Herzeliya Pituach, Israel), ciprofloxacin (Cayman Chemical Company, Ann Arbor, MI, USA), and chloramphenicol (Sigma, St. Louis, MO, USA). For comparison, a placebo varnish (SRV–placebo) was prepared in the same manner, except that no antibiotics were added to the formulation.

### 4.2. Doyle Nasal Silicone Splints

Sterile Doyle nasal silicone splints without an airway (65 mm × 25 mm × 1 mm; length × width × thickness) were purchased from Endure Industries, Inc. Homer, New York, NY, USA. The splints were cut into uniform 1 cm × 1 cm pieces for the experimental procedures outlined below. To ensure sterility, the splint segments were immersed in 70% ethanol for 1 h, followed by three washes with sterile double-distilled water (DDW), and then allowed to air dry before proceeding with the coating process.

### 4.3. Coating of Doyle Nasal Silicone Splints with SRV

The splint segments were coated with SRV–augmentin, SRV–ciprofloxacin, SRV–chloramphenicol or SRV–placebo by immersing them in the respective solutions and then allowed to air dry at room temperature until a fine film was formed. The coating process was repeated three times for each splint, with a 24 h interval between each application. The amount of SRV film formed on each piece was 7 ± 4 mg. The effects of SRV–augmentin-, SRV–ciprofloxacin-, and SRV–chloramphenicol-coated splints were compared with those of SRV–placebo-coated splints. All experiments were performed in triplicates.

### 4.4. Cultivation of Staphylococcus aureus

The bacterial model strain was *S. aureus* ATCC 25923, which can form biofilms on Doyle nasal silicon splints. A starter culture was prepared by inoculating 100 µL of a bacterial stock into 10 mL of tryptic soy broth (TSB) (Acumedia, Neogen, Lansing, MI, USA) and incubating overnight at 37 °C. The bacterial stock was kept at −80 °C in 15% glycerol in TSB. Fresh bacterial starter cultures were prepared daily. The overnight culture was diluted to an optical density (OD) of 0.1 at 600 nm in TSB and used for the experiments described below.

### 4.5. In Vitro Model for Studying the Long-Term Antibacterial and Antibiofilm Properties of SRV-Coated Splint Segments

SRV-coated splint segments were inserted into 24 flat-bottomed tissue-grade well plates (Corning Incorporation, Kennebunk, ME, USA), each well containing 1 mL of bacterial suspension in TSB at an initial OD_600nm_ of 0.1, followed by a 24 h incubation at 37 °C [54]. The Doyle segments were transferred daily to 1 mL of new, fresh bacterial cultures in TSB for another 24 h incubation. This procedure was performed for 22 days.

#### 4.5.1. Measuring the Effect of SRV-Coated Splint Segments on Planktonic Bacterial Growth

The turbidity of the planktonic growth phase was measured daily by recording the absorbance at 600 nm using a plate reader (Thermo Scientific Multiskan SkyHigh, Life Technologies, Holdings Pte Ltd., Singapore) [54]. The turbidity of bacterial samples exposed to SRV–antibiotic-coated splints was compared to those exposed to SRV–placebo-coated splints and to bacterial samples without splints. The absorbance of TSB without bacteria served as the background reads. To assess bacterial viability, the drop plate method was used: 10 μL of each sample was spotted on tryptic soy agar (TSA) plates followed by a 24 h incubation at 37 °C.

#### 4.5.2. Biofilm Metabolic Assay

The amount of biofilm formed on the well surface after incubating the bacteria in the presence of the SRV-coated stents was determined by MTT metabolic assay by incubating the PBS-washed biofilms with 0.5 mg/mL MTT (3-(4,5-Dimethylthiazol-2-yl)-2,5-diphenyltetrazolium bromide; Sigma, St. Louis, MO, USA) in PBS for 1 h at 37 °C [54]. The formazan formed was dissolved in 1 mL dimethylsulfoxide (DMSO), and the absorbance was read at 570 nm. The formula used for calculations was:% Metabolic activity = ((OD_Sample_ − OD_Background_)/(OD_Control_ − OD_Background_)) × 100%

### 4.6. Kinetic Disk Diffusion Assay to Determine the Long-Term Anti-Bacterial Activity of SRV-Coated Splints

SRV-coated splints were transferred daily to new TSA plates coated with 100 μL of an overnight bacterial culture of *S. aureus*, followed by a 24 h incubation at 37 °C [38]. The zone of inhibition was calculated by the clearance area (cm^2^) around the splint (r_1_ × r_2_ × π) after subtracting the area of the splints, where r_1_ and r_2_ are the radii of the vertical and horizontal directions, respectively. The assay was conducted until a steady reduction in the inhibition zone radius was achieved for SRV–chloramphenicol-coated splints, and until complete loss of detectable inhibitory activity was observed for splints coated with SRV–augmentin or SRV–ciprofloxacin.

### 4.7. High-Resolution-Scanning Electron Microscope (HR-SEM)

Biofilms formed on SRV-coated splints were visualized by HR-SEM. The splints were washed twice with DDW before fixation with 4% glutaraldehyde (Electron Microscopy Sciences, Hatfield, PA, USA) in DDW for 2 h. After fixation, the samples were washed with DDW and dried before coating with iridium and visualized by high-resolution scanning electron microscope (Apreo 2 S LoVac, Thermo Fisher Scientific Inc, Waltham, MA, USA) at various magnifications (×1000–×20,000) [54].

### 4.8. Statistical Analysis

Means of three or four independent experiments performed in at least triplicate were calculated. Statistical analysis was performed using the Student’s *t*-test in the Microsoft Excel Software, with a *p*-value < 0.05 considered statistically significant.

## 5. Conclusions

This study demonstrates that coating Doyle nasal silicone splints with sustained-release varnishes (SRVs) containing various antibacterial agents, such as augmentin, ciprofloxacin, or chloramphenicol, provides potent and long-lasting antibacterial protection against *Staphylococcus aureus* growth and biofilm formation. Among the tested agents, SRV–chloramphenicol proved the most effective, maintaining antibacterial activity for more than 65 days, followed by ciprofloxacin for 52 ± 1 days. In contrast, the antibacterial effect of SRV–augmentin-coated splints was limited to 14 ± 1 days, likely due to its inherent instability. These findings suggest that SRV coatings can convert passive splints into active drug-eluting devices, reducing the reliance on systemic antibiotics, improving patient compliance, and lowering the risk of antibiotic resistance.

Future in vivo studies are needed to validate these promising in vitro results and assess their clinical relevance. If successful, SRV-coated splints could represent a significant advancement in preventing postoperative infections and restenosis in sinonasal surgeries, with potential broader implications in other otorhinolaryngologic procedures.

## Figures and Tables

**Figure 1 pharmaceuticals-18-01746-f001:**
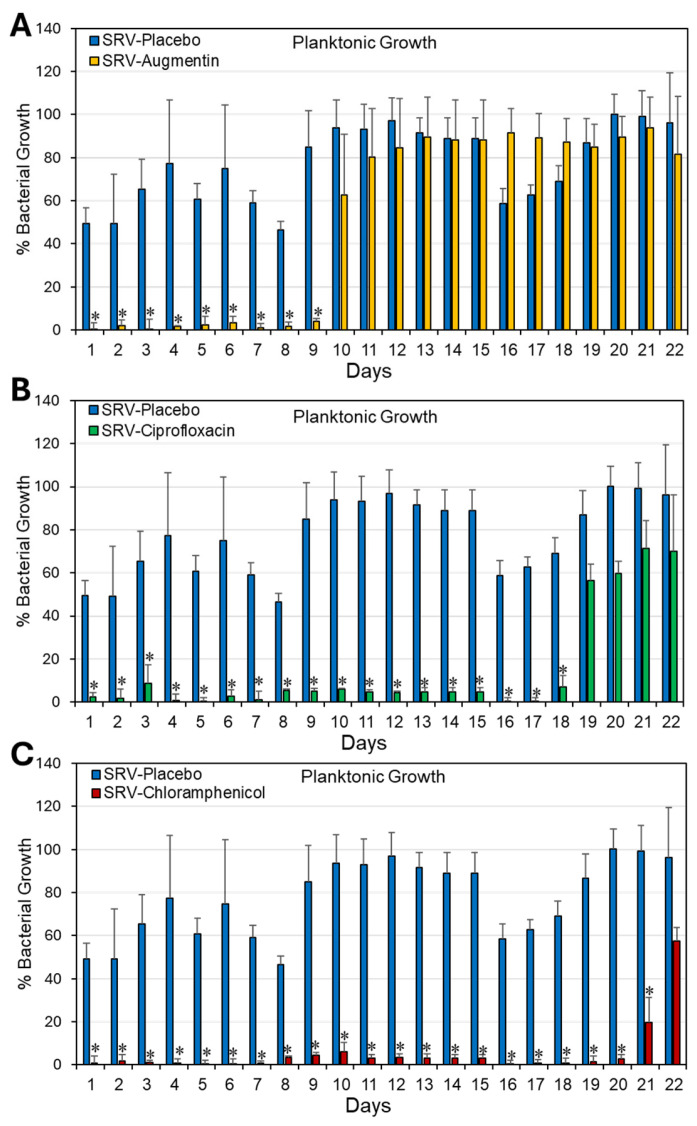
(**A**–**C**) Antibacterial effect of Doyle splints coated with SRV–augmentin (**A**), SRV–ciprofloxacin (**B**), or SRV–chloramphenicol (**C**) on *S. aureus* ATCC 25923. The antibacterial effect was determined by measuring the OD at 600 nm of planktonic growth. Calculations were performed relative to bacterial samples without splints, which were set to 100%. N = 4. * *p* < 0.05 compared to SRV–placebo-coated stents.

**Figure 2 pharmaceuticals-18-01746-f002:**
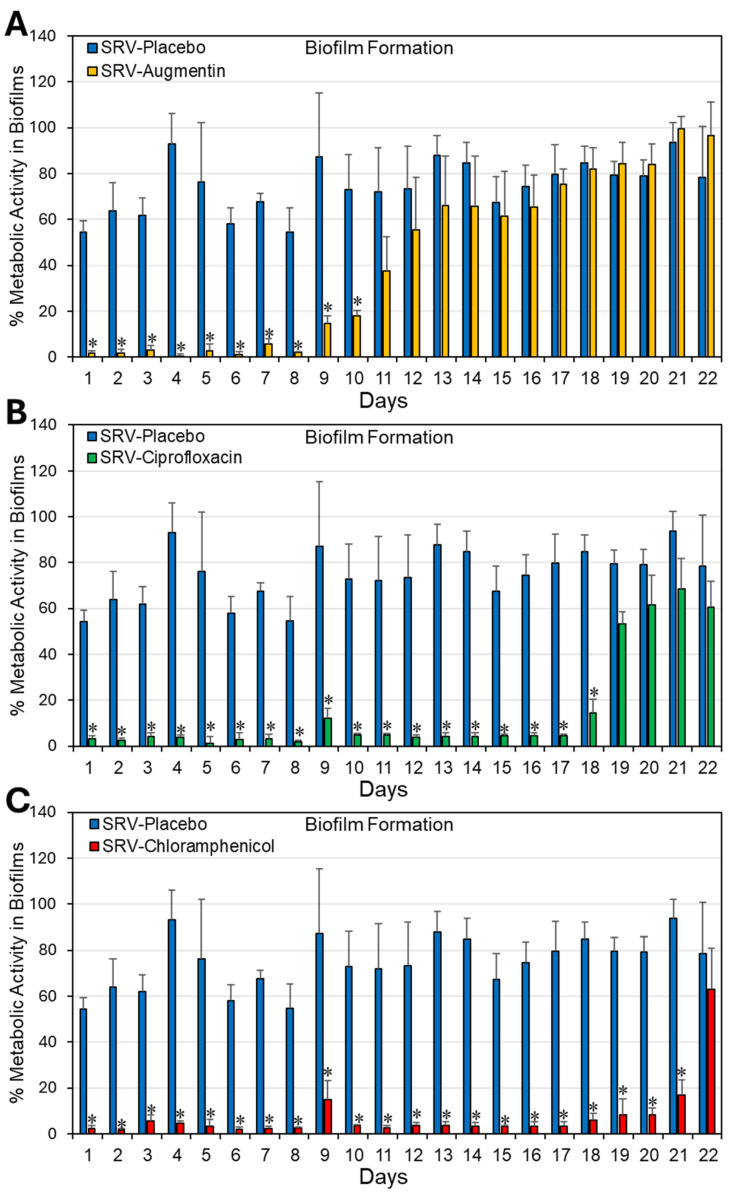
(**A**–**C**) Antibiofilm effect of Doyle splints coated with SRV–augmentin (**A**), SRV–ciprofloxacin (**B**), or SRV–chloramphenicol (**C**) on *S. aureus* ATCC 25923. The antibiofilm effect was determined by measuring the metabolic activity of the biofilms using MTT. The calculations were done against bacterial samples without splints, which were set to 100%. N = 4. * *p* < 0.05 compared to SRV–placebo-coated stents.

**Figure 3 pharmaceuticals-18-01746-f003:**
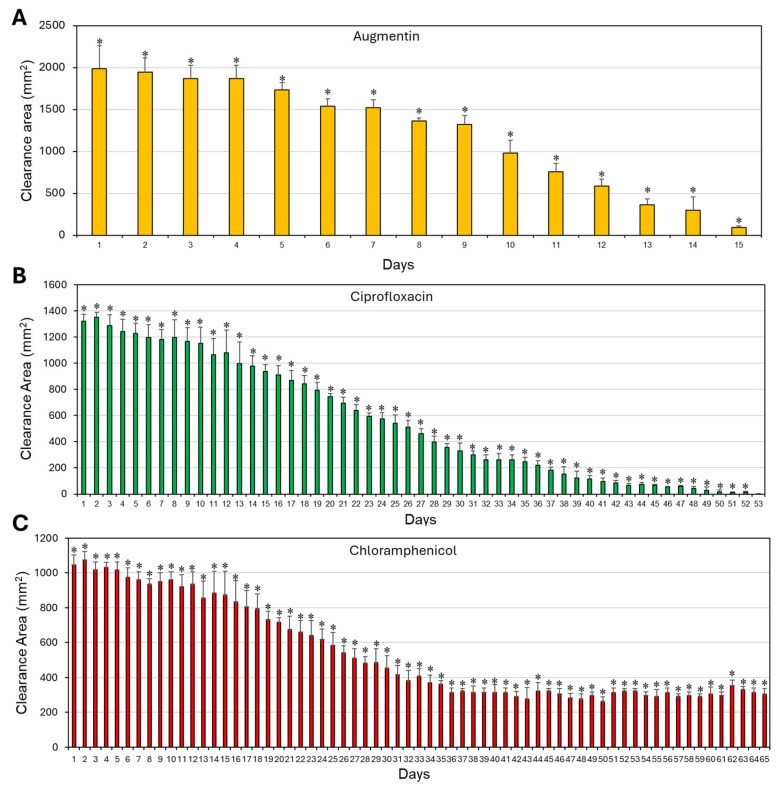
Long-term antibacterial activity of Doyle splint segments coated with SRV–antibiotics. (**A**–**C**). The clearance area of *S. aureus* 25923 around 1 cm × 1 cm splint segments coated with SRV–augmentin (**A**), SRV–ciprofloxacin (**B**), or SRV–chloramphenicol (**C**) after the indicated number of exposures to *S. aureus*. N = 3. * *p* < 0.05 compared to SRV–placebo-coated stents.

**Figure 4 pharmaceuticals-18-01746-f004:**
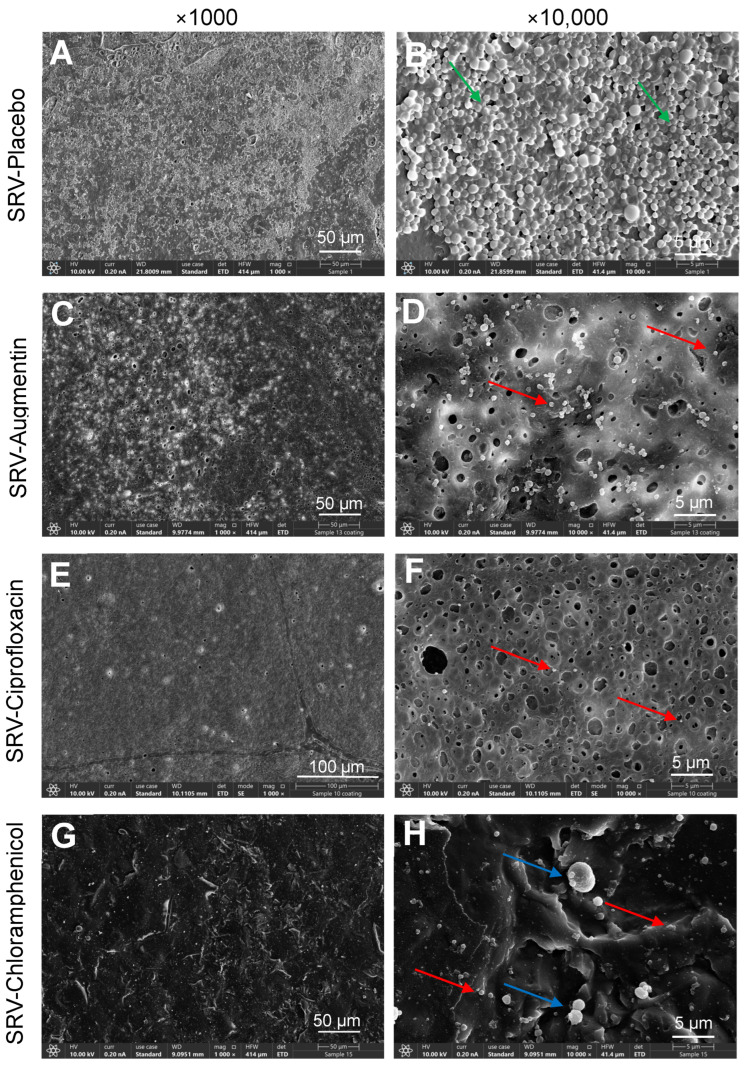
(**A**–**H**). HR-SEM images of Doyle nasal silicon splint segments coated with SRV–placebo (**A**,**B**), SRV–augmentin (**C**,**D**), SRV–ciprofloxacin (**E**,**F**), or SRV–chloramphenicol (**G**,**H**) that have been exposed 10 times to *S. aureus* 25923. Green arrow points to live bacteria, which appear as luminous, round 3D structures with smooth membranes, while red arrows point to the dead bacteria, which appear as smaller, irregular cells. Blue arrows point to enlarged, swollen bacteria. Magnifications: ×1000 (**A**,**C**,**E**,**G**) and ×10,000 (**B**,**D**,**F**,**H**). HR-SEM imaging was performed blindly with numbered samples without knowing the sample identities.

**Table 1 pharmaceuticals-18-01746-t001:** The composition of the SRV formulations used to coat Doyle nasal silicon splint.

Component	Placebo	Augmentin	Ciprofloxacin	Chloramphenicol
Ethylcellulose	1000 mg	1000 mg	1000 mg	1000 mg
Hydroxypropylcellulose	200 mg	200 mg	200 mg	200 mg
PEG-400	120 mg	120 mg	120 mg	120 mg
Active ingredient	None	600 mg Augmentin	600 mg Ciprofloxacin	600 mg Chloramphenicol
Ethanol	12 mL	12 mL	12 mL	12 mL

## Data Availability

The original contributions presented in this study are included in the article. Further inquiries can be directed to the corresponding author.

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
