# Peer review of "Coating Doyle Nasal Silicone Splints with a Sustained Release Varnish Containing Antibiotics Provides Long-Term Protection from Staphylococcus aureus: An In Vitro Study"

_pharmaceuticals, 2025, doi:10.3390/ph18111746_

Round 1

Reviewer 1 Report

Comments and Suggestions for Authors

The presented manuscript is dedicated to the antibacterial coating of Doyle nasal silicone splints. Overgrowth of Doyle's silicone nasal splint by biofilms is of practical significance in medicine, and the authors are working to solve this problem. This work is certainly useful and interesting. I have a few questions listed below:

  1. I would like to see a more detailed analysis of the sustained release varnish (SRV) method. This method has been extensively used in scientific research and has been proposed for suppressing biofilm formation for many years (https://doi.org/10.1089/end.2011.0140). However,is its practical application known in clinics? An example of such work would be helpful in the introduction, if the authors could find one. If no such examples were available, it would be useful to describe the limitations of this method and explain why it has not yet been widely adopted.
  2. Why did the authors study the metabolic analysis of biofilm in plate wells? It seems logical to conduct the MTT assay directly on varnished material samples to assess surface growth, washing them with PBS and transferring them to new wells. This is precisely what is required in practice, rather than suppressing biofilm formation on other surfaces around Doyle splints. Additionally, biofilm formation on the surface may be assessed using crystal violet staining. Measuring the amount of biofilm on the samples'surface is more suitable for this study than suppressing itsformation at a distance from the samples in the plate wells.
  3. The authors used Polyethylene glycol 400 (PEG-400, Sigma) as a varnish base. PEG-400 and chloramphenicol are known to be highly soluble in ethanol. Why did the authors choose ethanol sterilization as a method for their samples? This method of sterilizing samples (immersed in 70% ethanol for 1 hour) loses some of the varnish and some of the antibacterial agent. Why not use shorter exposure times or UV irradiation?
  4. Figures 2 and 3 show that SRV-placebo-coated stents inhibit planktonic growth and biofilm formation by 60% in the first few days. Please explain why this might be the case.
  5. In section 2.1, Latin names are not italicized.

Author Response

Point-to-point responses to Reviewer 1

We want to thank the reviewer for taking the time to critically review our manuscript.

The presented manuscript is dedicated to the antibacterial coating of Doyle nasal silicone splints. Overgrowth of Doyle's silicone nasal splint by biofilms is of practical significance in medicine, and the authors are working to solve this problem. This work is certainly useful and interesting. I have a few questions listed below:

Comment 1: I would like to see a more detailed analysis of the sustained release varnish (SRV) method. This method has been extensively used in scientific research and has been proposed for suppressing biofilm formation for many years (https://doi.org/10.1089/end.2011.0140). However, is its practical application known in clinics? An example of such work would be helpful in the introduction, if the authors could find one. If no such examples were available, it would be useful to describe the limitations of this method and explain why it has not yet been widely adopted.

Response 1: The study is at the preclinical stage to evaluate the feasibility of an SRV-antibiotic technology to be applied on Doyle's silicon nasal splint for long-term prevention of S. aureus infection. Similar SRV techniques have previously been used to incorporate other antibacterial and antifungal drugs, and an in vivo study in human has shown the efficacy of SRV-clotrimazole to prevent and reduce Candida albicans infection in the oral cavity. The transition from lab work to the clinical stage requires several clinical trials which are out of scope of the current research, although this has been initiated for other applications such as prevention of urinary tract infections by coating catheters with antibacterial SRV, which was tested on dogs. We have added the following text to the Introduction to better emphasize this issue:

"Our study is a preclinical in vitro investigation, and future research is required to support the transition of this platform into clinical use. SRV technology has previously been investigated in vitro in various applications aimed at preventing microbial infections through the incorporation of different antiseptic and antimicrobial compounds, including chlorhexidine [34,35], triclosan [36], ciprofloxacin [37], and clotrimazole [38]. In vivo studies have further demonstrated the clinical potential of this approach: Coating catheters with SRV-chlorhexidine reduced catheter-associated biofilms in dogs [39], while application of SRV-clotrimazole in human volunteers decreased Candida albicans load in the oral cavity [40]."

Comment 2: Why did the authors study the metabolic analysis of biofilm in plate wells? It seems logical to conduct the MTT assay directly on varnished material samples to assess surface growth, washing them with PBS and transferring them to new wells. This is precisely what is required in practice, rather than suppressing biofilm formation on other surfaces around Doyle splints. Additionally, biofilm formation on the surface may be assessed using crystal violet staining. Measuring the amount of biofilm on the samples' surface is more suitable for this study than suppressing its formation at a distance from the samples in the plate wells.

Response 2: We agree with the Reviewer that we could have done MTT metabolic assay on the coated Doyle segments. Our assumption was that if the amount of antimicrobial agent released from the coatings was sufficient to prevent bacterial growth and biofilm formation in its surroundings, it would also be sufficient to prevent bacterial growth and biofilm formation on the coated segments, which has even a higher compound level than the surrounding fluid. The lack of bacteria on the coated Doyle surfaces was demonstrated by HR-SEM (Figure 4). We have added the following text to the Result section to better emphasize this issue:

"Our assumption was that if the amount of antimicrobial agent released from the coatings was sufficient to inhibit bacterial growth and biofilm formation in the surrounding medium, it would likewise be sufficient to prevent bacterial colonization and biofilm formation on the coated segments themselves, where the local concentration of the compound is even higher."

Comment 3: The authors used Polyethylene glycol 400 (PEG-400, Sigma) as a varnish base. PEG-400 and chloramphenicol are known to be highly soluble in ethanol. Why did the authors choose ethanol sterilization as a method for their samples? This method of sterilizing samples (immersed in 70% ethanol for 1 hour) loses some of the varnish and some of the antibacterial agent. Why not use shorter exposure times or UV irradiation?

Response 3: The Doyle segments were sterilized with 70% ethanol prior to coating, not after coating which would result in the dissolution of the varnish. 70% Ethanol is sufficient for sterilizing the segments.

Comment 4: Figures 2 and 3 show that SRV-placebo-coated stents inhibit planktonic growth and biofilm formation by 60% in the first few days. Please explain why this might be the case.

Response 4: The observed decrease in planktonic growth and biofilm formation by SRV-placebo-coated segments may reflect a physical or surface-related interference effect arising from the presence of the segments themselves, rather than from any antimicrobial activity. This explanation has now been added to the Result section.

Comment 5: In section 2.1, Latin names are not italicized.

Response 5: According to our understanding, bacterial names should be in italics. We submitted the bacterial name in italics, and seemingly through the processing in the MDPI site, they become in regular letters. We have corrected these to italics.

Reviewer 2 Report

Comments and Suggestions for Authors

The submitted manuscript presents a well-conceived and carefully executed study addressing a highly relevant and topical issue — the prevention of persistent infections associated with medical implants. Recurrent infections caused by pathogenic or opportunistic microorganisms represent a major clinical challenge, and the proposed approach based on sustained-release varnish (SRV) coatings containing antibiotics is both innovative and of potential translational value.

The study is well-structured, the Introduction section is informative and effectively contextualizes the research problem, and the experimental design is generally sound. I particularly appreciate that the authors did not limit their investigation to planktonic bacterial cells but also included biofilm formation, which adds considerable depth and scientific value to the work.

The results are clearly presented and supported by an adequate number of experiments. The discussion appropriately interprets the data and highlights the antibacterial efficacy of the coatings tested. Although several textual inconsistencies and minor methodological ambiguities are present, these issues are relatively minor and can be corrected easily. Overall, I consider this manuscript to be of good quality and suitable for publication after minor revisions.

Specific Comments

  1. Italicization of Latin names – Please ensure consistent use of italics for Latin bacterial names throughout the text (e.g., Staphylococcus aureus), see for instance Lines 132, 133, 137, 138, etc.

  2. Line 139 – "no effect" – The description “no effect” should be clarified or expanded. Please specify what parameters were unaffected and to what extent.

  3. Figure 1 – I recommend unifying the naming format of figures. Remove “A–C” from the title and instead specify each subpanel in the caption (e.g., “with SRV-augmentin (A), SRV-ciprofloxacin (B), and SRV-chloramphenicol (C)”). The caption currently lacks this detailed specification.

  4. Figure 1 title – Remove the redundant full stop at the end of the figure title.

  5. Standard deviation anomalies – In Figure 1, the SD values for days 4 and 6 appear unusually large compared to adjacent time points (e.g., day 5). Please verify the correctness of these calculations.

  6. Figure 2 – Apply the same formatting and structural improvements as suggested for Figure 1.

  7. Figure 3 – The first image in this figure is of lower quality and should be improved for clarity. In addition, please ensure that panels A, B, and C are clearly labeled and referenced consistently in both the figure and caption.

  8. Line 260 – The manufacturer’s name is incomplete; ensure that full company details are provided and formatting is consistent across the text.

  9. Lines 281–282 – The methodological description of the “frozen bacterial stock” handling is unclear. It seems unlikely that it was pipetted directly; please revise this section for clarity.

  10. Line 291 – The methodological description in this line is again somewhat ambiguous; I recommend rephrasing and expanding this part to make it fully understandable.

Author Response

Point-to-point response to Reviewer 2

We want to thank the reviewer for taking the time to critically review our manuscript.

The submitted manuscript presents a well-conceived and carefully executed study addressing a highly relevant and topical issue — the prevention of persistent infections associated with medical implants. Recurrent infections caused by pathogenic or opportunistic microorganisms represent a major clinical challenge, and the proposed approach based on sustained-release varnish (SRV) coatings containing antibiotics is both innovative and of potential translational value.

The study is well-structured, the Introduction section is informative and effectively contextualizes the research problem, and the experimental design is generally sound. I particularly appreciate that the authors did not limit their investigation to planktonic bacterial cells but also included biofilm formation, which adds considerable depth and scientific value to the work.

The results are clearly presented and supported by an adequate number of experiments. The discussion appropriately interprets the data and highlights the antibacterial efficacy of the coatings tested. Although several textual inconsistencies and minor methodological ambiguities are present, these issues are relatively minor and can be corrected easily. Overall, I consider this manuscript to be of good quality and suitable for publication after minor revisions.

Specific Comments

Comment 1: Italicization of Latin names – Please ensure consistent use of italics for Latin bacterial names throughout the text (e.g., Staphylococcus aureus), see for instance Lines 132, 133, 137, 138, etc.

Response 1: We submitted the manuscript with bacterial names in italics. However, it seems that the MDPI editorial has changed these to regular letters.  We have corrected it back to italics.

Comment 2: Line 139 – "no effect" – The description “no effect” should be clarified or expanded. Please specify what parameters were unaffected and to what extent.

Response 2: We have accordingly corrected the sentence to: "As expected, SRV-placebo-coated splints had no significant antibacterial or anti-biofilm effect".

Comment 3: Figure 1 – I recommend unifying the naming format of figures. Remove “A–C” from the title and instead specify each subpanel in the caption (e.g., “with SRV-augmentin (A), SRV-ciprofloxacin (B), and SRV-chloramphenicol (C)”). The caption currently lacks this detailed specification.

Response 3: The changes have now been made in the figure legends.

Comment 4: Figure 1 title – Remove the redundant full stop at the end of the figure title.

Response 4: The redundant full stop was removed from the title.

Comment 5: Standard deviation anomalies – In Figure 1, the SD values for days 4 and 6 appear unusually large compared to adjacent time points (e.g., day 5). Please verify the correctness of these calculations.

Response 5: We have rechecked the data, and these are the SD values of four representative replicates.

Comment 6: Figure 2 – Apply the same formatting and structural improvements as suggested for Figure 1.

Response 6: Corrections done.

Comment 7: Figure 3 – The first image in this figure is of lower quality and should be improved for clarity. In addition, please ensure that panels A, B, and C are clearly labeled and referenced consistently in both the figure and caption.

Response 7: The A-C letters have been added to the figure, and some of the letters were in grey, and now corrected to black.

Comment 8: Line 260 – The manufacturer’s name is incomplete; ensure that full company details are provided and formatting is consistent across the text.

Response 8: This has now been corrected.

Comment 9: Lines 281–282 – The methodological description of the “frozen bacterial stock” handling is unclear. It seems unlikely that it was pipetted directly; please revise this section for clarity.

Response 9: For clarification, we have deleted "frozen" in the sentence and added the following text: "The bacterial stock was kept at – 80 °C in 15% glycerol in TSB."

Comment 10: Line 291 – The methodological description in this line is again somewhat ambiguous; I recommend rephrasing and expanding this part to make it fully understandable.

Response 10: We have rephased the text: "The Doyle segments were transferred daily to 1 mL of new, fresh bacterial cultures in TSB for another 24 h incubation. This procedure was performed for 22 days."

Round 2

Reviewer 1 Report

Comments and Suggestions for Authors

The submitted manuscript is ready for publication in its current form.